# Genomic Insights into the Molecular Basis of Broad Host Adaptability of the Entomopathogenic Fungus *Conidiobolus coronatus* (Entomophthoromycotina)

**DOI:** 10.3390/jof11080600

**Published:** 2025-08-19

**Authors:** Fan Bai, Tian Yang, Lvhao Zhang, Jiaqi Yang, Xinyu Chen, Xiang Zhou

**Affiliations:** National Key Laboratory for Development and Utilization of Forest Food Resources, School of Forestry and Biotechnology, Zhejiang A&F University, Hangzhou 311300, China; fbai@stu.zafu.edu.cn (F.B.); 2021102032004@stu.zafu.edu.cn (T.Y.); 2020102081016@stu.zafu.edu.cn (L.Z.); jqyang@stu.zafu.edu.cn (J.Y.); xychen@stu.zafu.edu.cn (X.C.)

**Keywords:** insect pathogenic fungi, generalist pathogen, genomic profiling, metabolomic profiling, genome assembly

## Abstract

*Conidiobolus coronatus* (Entomophthorales), a fungal pathogen with a broad insect host range, is a promising candidate for biocontrol applications. We sequenced a *C. coronatus* strain isolated from a *Rhopalomyia* sp. cadaver using PacBio long-read sequencing to elucidate the molecular basis of its wide host adaptability. The newly assembled 44.21 Mb genome exhibits high completeness (BUSCO score: 93.45%) and encodes 11,128 protein-coding genes, with 23.1% predicted to mediate pathogen–host interactions. Comparative genomics with the aphid-obligate pathogen *C. obscurus* revealed significant expansions in gene families associated with host adaptation mechanisms, including host recognition, transcriptional regulation, degradation of host components, detoxification, and immune evasion. Functional annotation highlighted enrichment in cellular component organization and energy metabolism. Pfam annotation identified one hundred twenty-five seven-transmembrane receptors (putative GPCRs), sixty-seven fungus-specific transcription factors, three hundred sixty-one peptidases (one hundred ninety-eight serine proteases and one hundred three metalloproteases), one hundred twenty-seven cytochrome P450 monooxygenases (P450s), thirty-five cysteine-rich secretory proteins, and fifty-five tyrosinases. Additionally, four hundred thirty carbohydrate-active enzymes (CAZymes) across six major modules were characterized. Untargeted metabolomics detected 22 highly expressed terpenoids, consistent with terpenoid biosynthesis gene clusters in the genome. Collectively, these expansions underpin the broad host range of *C. coronatus* by enabling cross-host signal decoding and gene expression reprogramming, breaching diverse host physicochemical barriers, and expanding its chemical ecological niche. This study provides genomic insights into broad host adaptability in entomopathogenic fungi, facilitating further understanding of pathogen–host interactions.

## 1. Introduction

Entomopathogenic fungi, which invade insects by direct penetration of the host cuticle, serve as crucial natural regulators of insect populations [1]. Species from the genera *Beauveria* and *Metarhizium*, such as *Beauveria bassiana* and *Metarhizium anisopliae*, have been developed as commercial biopesticides for agricultural and forestry pest control [2]. Leveraging genomic advances, genetic modification can further enhance the efficacy of fungal insecticides [3]. For instance, engineering UV tolerance in *Beauveria* isolates expands their applicability and field persistence without compromising inherent virulence [4,5]. Beyond genetic improvements, their broad host ranges (generalist strategy) constitute a key factor in commercial success [6,7]. Unlike hypocrealean fungi (Ascomycota), entomopathogens within the subphylum Entomophthoromycotina (Zoopagomycota) exhibit higher virulence, manipulate host behavior to their advantage, and can trigger insect epizootics [8,9,10]. However, most are host-specialized (specialists) with narrow host ranges, limiting their practical application. Notable exceptions include entomophthoralean species, like *Zoophthora radicans* (five insect orders; Entomophthoraceae) and *Conidiobolus coronatus* (six insect orders as well as arthropods outside Insecta; Ancylistaceae), which demonstrate broad adaptability [8,11,12,13]. Elucidating the molecular mechanisms underlying the broad host range is critical for harnessing their biocontrol potential [14].

Omic approaches provide powerful tools to dissect host-range divergence between generalist and specialist entomopathogens [7,15]. In Hypocreales, generalists typically adapt via genome expansion, characterized by proliferation of secreted proteases (e.g., subtilisin and trypsin), G-protein-coupled receptors (GPCRs), detoxifying enzymes (e.g., cytochrome P450 monooxygenases, P450s), and acquisition of novel gene clusters encoding nonselective toxins [3,15]. Specialists, conversely, restrict the host range through genome reduction and contraction of key gene families [15]. Genome size alone is not a reliable predictor of the host range due to substantial influences from repetitive sequences and transposon activity [16]. For example, within Entomophthorales, the specialist fly-obligate pathogen *Entomophthora muscae* possesses a genome exceeding 1 Gb [16]. Such giant genomes are prevalent in the family Entomophthoraceae, contrasting sharply with average 40 Mb genomes in Ancylistaceae, reflecting divergent evolutionary trajectories [14,16,17,18].

The *Conidiobolus* species encompass saprophytes (e.g., *C. heterosporus*), obligate pathogens (e.g., aphid-obligate *C. obscurus*), and generalist pathogens (e.g., *C. coronatus*) [14,17,18]. As basal fungi, *Conidiobolus* spp. are typically multinucleate, with predominantly haploid somatic nuclei [14]. However, heterozygosity complicates genome assembly, often resulting in comparatively low BUSCO (Benchmarking Universal Single-Copy Ortholog) completeness scores [16]. Furthermore, conspecific strains exhibit significant intraspecific genome size variation (10–20%) [19,20]. A strain was recently isolated from a naturally infected *Rhopalomyia* sp. gall midge cadaver within the campus of Zhejiang A&F University (Hangzhou, China; 30.26° N, 119.73° E), followed by monosporic purification. Morpho-molecular characterization identified this strain as *C. coronatus* (Figure 1). Beyond displaying diverse sporulation patterns, this strain exhibited potent pathogenicity against *Hyphantria cunea* larvae (Appendix A), confirming its broad-host-range trait. Pacbio Sequel II sequencing yielded a 44.2 Mb genome, larger than the 39.9 Mb reference genome of *C. coronatus* NRRL 28638. This study employs comparative genomics to elucidate the molecular basis underlying *C. coronatus*’s broad host adaptation, providing insights into Entomophthorales–host interactions.

## 2. Materials and Methods

### 2.1. Fungal Culture and DNA Extraction

The *C. coronatus* strain 20220622F was cultured on Sabouraud dextrose agar supplemented with yeast extract (40 g L^−1^ dextrose, 10 g L^−1^ peptone, 10 g L^−1^ yeast extract, and 15 g L^−1^ agar) in Petri dishes at 24 ± 1 °C with a 12:12 h light:dark photoperiod for 4 days. Culture pieces were mashed and transferred to 50 mL of liquid medium within a 150 mL flask, followed by incubation in a shaker at 120 rpm and 24 ± 1 °C for 3 days. Fresh mycelium from liquid culture was harvested, flash-frozen in liquid nitrogen, and pulverized into a fine powder for DNA extraction. The total genomic DNA was extracted using a TakaRa universal genomic DNA extraction kit (Tokyo, Japan) according to the manufacturer’s protocol. Purified DNA was subsequently sent to Biomarker Technologies (Beijing, China) for sequencing [18].

### 2.2. Genome Sequencing, Assembly, and Annotation

Whole-genome sequencing was performed on the Pacific Biosciences (PacBio) Sequel II platform (third-generation long-read sequencing) using circular consensus sequencing technology (CCS; https://github.com/PacificBiosciences/ccs, accessed on 30 July 2024), following the standard protocol [21]. Subreads were processed with the SMRT Link CCS program (parameters: min-passes, 5; min-rq, 0.9) to generate high-quality consensus reads [22]. The assembly was conducted using Hifiasm [23], followed by error-correction with Pilon using short-read sequencing data [24]. The genome assembly completeness was evaluated via alignment with BWA and BUSCO v2.0 against the Benchmarking Universal Single-Copy Ortholog database [25,26]. The assembled genome was deposited at GenBank under accession PRJNA1264516.

Due to the low conservation of repetitive sequences across species, constructing a species-specific repeat sequence database is essential for their accurate prediction. We built a repeat library using LTR_FINDER v1.05, MITE-Hunter, RepeatScout v1.0.5, and PILER-DF v2.4 [27,28,29,30]. This library was integrated with the Repbase database to form the final database [31]. Repeat sequences were then annotated using RepeatMasker v4.0.6 [32].

Gene prediction combined ab initio prediction- (Augustus v2.4, GlimmerHMM v3.0.4, GeneID v1.4, and SNAP) and homology-based methods (GeMoMa v1.3.1). Prediction results were integrated using EVidenceModeler (EVM) v1.1.1 [33]. Transfer RNA (tRNA) genes and ribosomal RNA (rRNA) genes were identified with tRNAscan-SE v1.3.1 and Infernal v1.1.1, respectively [34,35].

Functional annotation of protein-coding genes was performed by BLAST (v2.9.0) alignment (e-value cutoff value: 1 × 10^−5^) against KOG [36], KEGG [37], Swiss-Prot [38], Gene Ontology (GO) [39], Pfam [40], the Pathogen–Host Interaction database (PHI) [41], Carbohydrate-Active enZymes Database (CAZy) [42], and the Cytochrome P450 Engineering Database (CYPED) [43].

The protein sequences of all the predicted genes were analyzed using SignalP 4.0 to identify the proteins containing the signal peptides [44] and TMHMM to detect transmembrane domains [45]. Secreted proteins were identified as those containing predicted signal peptides but lacking transmembrane domains. For comparative analysis, we performed a re-annotation analysis of the *C. obscurus* strain ARSEF 7217 genome [18] using the methods described above.

### 2.3. Phylogenetics, Conserved Motifs, and Domain Analysis

Phylogenetic relationships for serine proteases, G-protein-coupled receptors (GPCRs), and fungus-specific transcription factors were inferred using MEGA v12 [46]. Maximum-likelihood phylogenetic trees were constructed under the Poisson correction model with 500 bootstrap replicates. Conserved motifs and domains were analyzed via MEME Suite (https://meme-suite.org, accessed on 30 August 2024) and CD-Search (https://www.ncbi.nlm.nih.gov/Structure/cdd/wrpsb.cgi, accessed on 30 August 2024), respectively.

### 2.4. Exploration of Terpene Metabolites in C. coronatus

Secondary-metabolite biosynthetic gene clusters (BGCs) were mined using antiSMASH v6.0 [47], identifying two terpenoid biosynthesis gene clusters. Untargeted metabolomic analysis was performed on 4-day liquid-cultured mycelia of *C. coronatus* 20220622F and *C. obscurus* ARSEF7217. Metabolites were extracted via standardized protocols and profiled using ultrahigh-performance liquid chromatography (UHPLC, Waters Acquity I-Class PLUS, Waters Corporation, Milford, MA, USA) coupled to a quadrupole time-of-flight high-resolution mass spectrometer (Q-TOF HRMS, Waters Xevo G2-XS, Waters Corporation, Milford, MA, USA) [48]. Raw data were processed using MassLynx (v4.2) for peak extraction and alignment. Metabolite identification was conducted using the METLIN and BMKGENE databases, supplemented by in silico fragment analysis. Following data preprocessing and quality control, differential expression analysis and functional annotation were conducted. High-dimensional data were analyzed using univariate (*t*-test) and multivariate (principal component analysis and orthogonal partial least squares discriminant analysis (OPLS-DA)) methods. Functional annotation utilized the KEGG database and LIPID Metabolites and Pathways Strategy (LIPID MAPS) for lipid classification. Annotated terpenoid compounds exhibiting significantly highly expression in *C. coronatus* were selected based on log_2_(Fold changes, FCs) ≥ 1, variable importance in projection ≥ 1 from OPLS-DA, and *p* < 0.05.

## 3. Results

### 3.1. Genomic Characteristics of C. coronatus

Whole-genome CCS was performed using the PacBio Sequel II platform, generating 3.84 Gb of CCS reads with an N50 length of 10,712 bp (Table 1; Appendix A). The *C. coronatus* genome assembly achieved a total length of 44.21 Mb at 86.85× coverage depth, comprising 77 scaffolds with an N50 length of 1,415,773 bp and a G + C content of 27.65% (Appendix A). BWA and BUSCO assessments demonstrated 99.91% genome coverage and 93.45% completeness (Appendix A). Repeat sequence analysis using the custom repeat library identified 7,745,087 bp of repetitive sequences, representing 17.52% of the genome (Appendix A).

A total of 11,128 protein-coding genes were predicted through ab initio- and homology-based approaches, with an average gene length of 1348 bp (Appendix A). Gene structure annotation revealed 31,664 exons and 20,536 introns (Appendix A). Additionally, 1631 non-coding RNAs (ncRNAs) spanning 67 families were annotated, including 294 rRNAs, 1294 tRNAs, and 43 other ncRNAs (Table 1).

Comparative analysis of publicly available entomophthoralean genomes revealed that the genome assembly quality of *C. coronatus* strain 20220622F significantly exceeded that of the first-reported reference strain, *C. coronatus* NRRL 28638 (GenBank accession no. GCA_001566745.1; released February 24, 2016). Key assembly metrics, including sequencing coverage, scaffold N50 length, and BUSCO completeness, demonstrated substantial improvements (Table 2). Notably, genomic features, such as the protein-coding gene count and G + C content, were evolutionarily conserved with those of the aphid-specialist *C. obscurus* ARSEF7217. Pronounced divergence was observed relative to the fly-obligate pathogen *E. muscae*, which exhibits extreme genomic expansion (38,917 genes) and a gigabase-scale genome (1.03 Gb).

### 3.2. General Functional Annotation of C. coronatus Genes

Among the 11,128 protein-coding genes identified in *C. coronatus*, 8025 genes (72.1%) received functional annotations through cross-database integration (Appendix A). Of these, 3702 genes were assigned GO terms (Figure 2A; Appendix A), distributed across molecular function (13 subcategories), cellular component (15 subcategories), and biological process (17 subcategories). The predominant GO terms were cell part (2205 genes), cell (2146), cellular process (2039), metabolic process (1801), and catalytic activity (1765). In contrast, *C. obscurus* exhibited significantly fewer GO annotations (2375 genes) [18], with its top terms being catalytic activity (1339), metabolic process (1074), and cellular process (1028). The expanded repertoire of cell-related genes in *C. coronatus* (e.g., cell: 2146 vs. 690 in *C. obscurus*) correlates with its enhanced capacity for conidial polymorphism. As shown in Figure 1E–H, *C. coronatus* produces diverse conidial forms, including microconidia and villose conidia, enabling adaptive fungal dispersal and survival.

Furthermore, three thousand seven hundred forty-two genes were annotated using the KEGG database (Figure 2B; Appendix A), covering metabolism (twenty-nine subcategories), genetic information processing (fifteen subcategories), cellular processes (five subcategories), and environmental information processing (one subcategory). The most enriched pathways in *C. coronatus* included oxidative phosphorylation (138 genes), ribosome (118), biosynthesis of amino acids (111), protein processing in endoplasmic reticula (114), and purine metabolism (109). For *C. obscurus*, the top enriched pathways were ribosome (150), protein processing in endoplasmic reticula (133), RNA transport (123), biosynthesis of amino acids (123), and oxidative phosphorylation (112). Additionally, 5571 genes of *C. coronatus* were annotated with KOG functional categories, fewer than the 6465 annotated in *C. obscurus* (Appendix A). The most prominent function contrasts between the two species were observed in energy production and conversion (391 vs. 339 genes), amino acid transport and metabolism (369 vs. 349 genes), and secondary-metabolite biosynthesis, transport, and catabolism (262 vs. 237 genes). In integrated KEGG and KOG analyses, *C. coronatus* exhibits significant advantages in energy metabolism (particularly oxidative phosphorylation) and secondary-metabolite biosynthesis, while also demonstrating strong amino acid metabolic capabilities.

### 3.3. Special Functional Annotation of C. coronatus Genes

#### 3.3.1. CAZy Database Annotation

The CAZy database (http://www.cazy.org/, accessed on 30 August 2024) catalogs enzymes involved in glycosidic bond degradation, modification, and biosynthesis [42]. Analysis identified four hundred thirty CAZymes in *C. coronatus*, comprising five major enzyme classes alongside carbohydrate-binding modules (Figure 3): glycoside hydrolases (GHs), one hundred one enzymes (nineteen families, 23.48%); glycosyl transferases (GTs), one hundred forty-one enzymes (twenty-seven families, 32.79%); polysaccharide lyases (PLs), one enzyme (one family, 0.23%); carbohydrate esterases (CEs), eighty-five enzymes (nine families, 19.76%); auxiliary activities (AAs), seventy-two enzymes (seven families, 16.74%); and thirty carbohydrate-binding modules (CBMs, nine families, 6.97%). *C. coronatus* exhibits significant expansion in the GT1, GT71, CE10, AA7, and AA11 families compared to *C. obscurus*, which displayed high diversity in the GH18 (36 genes) and CBM19 (11 genes) families (Appendix A).

#### 3.3.2. PHI-Base Annotations

PHI-base curates experimentally validated or literature-reported pathogenic genes, virulence factors, and effector proteins from bacterial, fungal, and other pathogens infect plants, fungi, or insects [41]. Based on the *C. coronatus* genome, 2575 genes were annotated in PHI-base, with 73.2% predicted to associate with fungal pathogenicity (Figure 4A). These included 1118 genes (43.4%) linked to reduced virulence (weakening pathogenicity), 74 genes (2.9%) linked to increased virulence, and 99 genes (3.8%) linked to host lethality. Compared to *C. obscurus*, *C. coronatus* possessed more genes related to mixed outcomes (240 vs. 65 genes) and effectors (88 vs. 73 genes), whereas *C. obscurus* had higher numbers of genes associated with increased virulence (110 vs. 11 genes) and lethality (136 vs. 99 genes).

Further analysis of PHI-annotated genes in *C. coronatus* via KOG classification revealed significant enrichment within the O functional category (Posttranslational modification, protein turnover, and chaperones), accounting for 32.7% of all the PHI-associated genes (Figure 4B). This enrichment probably reflects adaptations in the secretory virulence machinery. We identified 1316 secreted proteins harboring signal peptides but lacking transmembrane domains. Functional classification highlighted their roles in the host’s barrier degradation and immune suppression, especially exhibiting expansions in metallopeptidases M28 and M36 (twenty-three vs. zero genes in *C. obscurus*), tyrosinase (fifty-five vs. twenty-four), and cysteine-rich secretory proteins (twenty-three vs. fifteen) (Appendix A).

#### 3.3.3. Protein Family (Pfam) Domain Annotation

Comparative Pfam domain analysis reveals that *C. coronatus* exhibits distinct protein family features associated with host adaptation (Figure 5). Notably, *C. coronatus* possesses more genes encoding seven-transmembrane receptors (PF00001.16; one hundred twenty-five vs. twenty-eight genes in *C. obscurus*), fungus-specific transcription factors (PF04082.13; sixty-seven vs. twenty-five), glutathione S-transferase N-terminal domains (PF13417.1; forty-three vs. eleven), winged-helix–turn-helix DNA-binding domains (PF13412.1; forty-two vs. one), and alpha/beta-hydrolase folds (PF07859.8; thirty-nine vs. fifteen). Meanwhile, *C. coronatus* shows comparable abundance in P450s (PF00067.17; 127 vs. 124). Integrated CYPED analysis further identifies CYP51 as the most abundant subfamily (Appendix A). These findings suggest that *C. coronatus* has evolved enhanced capabilities for environmental sensing, xenobiotic detoxification, and transcriptional regulation, probably supporting its adaptive flexibility in diverse ecological niches.

### 3.4. Abundant Serine Proteases and Metallopeptidases in C. coronatus

Proteolytic enzymes play a pivotal role in fungal insect infections, where degradation of insect epidermal components requires the synergistic action of diverse peptidases, directly influencing the host range specificity. Generalist entomopathogenic fungi are reported to encode more peptidases than specialists [3]. Pfam annotation identified one hundred ninety-eight serine protease genes in *C. coronatus*, distributed across ten families: trypsin (S1A, ninety genes), subtilisin (S8, sixty), prolyl oligopeptidase (S9, eighteen), carboxypeptidase Y (S10, twelve), lon protease (S16, two), signal peptidase I (S26, two), acid prolyl endopeptidase (S28, eight), rhomboid (S54, four), nucleoporin (S59, one), and LD-carboxypeptidase (S66, one) (Figure 6A). Compared to *C. obscurus*, *C. coronatus* exhibited a higher number of genes within trypsin (S1A) and subtilisin (S8) families (Appendix A).

MEME-based motif analysis confirmed conserved sequences flanking the catalytic triad residues (DTG, GHGTH, and SGTS) in subtilisin of both species (Figure 6C). Likewise, trypsin exhibited conserved active-site residues (His, Asp, and Ser; HDS motif) with highly analogous motif architectures (Figure 6B).

The *C. coronatus* genome also displays significant expansion in metallopeptidase genes (107 vs. 78 in *C. obscurus*, Figure 6D). Notably, it exhibits an increased number of M20/M25/M40 domains (fourteen vs. five, Figure 6E; PF01546.23), which cleave zinc-dependent peptide bonds at N-termini, suggesting their expansion facilitates adaptive proteolysis across diverse host substrates. Additionally, *C. coronatus* possesses a markedly expanded repertoire of M36 metallopeptidase genes (fourteen vs. one).

### 3.5. Expansion of G-Protein-Coupled Receptors in C. coronatus

Fungal GPCRs mediate host recognition and activate downstream signaling pathways critical for infection initiation [49]. Pfam database annotation identified 125 GPCR-encoding genes in the *C. coronatus* genome, significantly exceeding the 28 observed in *C. obscurus*. This expansion suggests that broad-host-range entomopathogenic fungi employ increased GPCR diversity to detect heterogeneous host environmental signals.

Further analysis of the 125 GPCRs in *C. coronatus*, using MEME suite, revealed 10 conserved motifs (Figure 7A). Among these, motif 10 exhibited high conservation, with an aspartic acid residue present in 123 GPCRs (Figure 7B). Domain annotation classified one hundred twelve functionally annotated GPCRs into three categories: seven tm class-A rhodopsin-like, seven tm GPCR superfamily, and seven tm 1 superfamily.

### 3.6. Expansion of Transcription Factors in C. coronatus

Pfam annotation identified diverse transcription factors in *C. coronatus* (Appendix A), including fungus-specific transcription factor domain, bZIP transcription factor, histone-like transcription factor, fungal Zn2-Cys6 binuclear cluster domain, and zinc-finger double domain. This repertoire highlights the complexity of its transcriptional network. Compared to *C. obscurus*, *C. coronatus* contains 67 genes encoding the fungus-specific transcription factor domain (Figure 8). Among these, 11 transcription factors were annotated using the Swiss-Prot database, with most involved in regulating nitrogen assimilation processes (Appendix A). This abundance suggests their role as core regulatory elements in *C. coronatus* responses to environmental signals, potentially facilitating its adaptation to multiple hosts.

### 3.7. Metabolomic Profiling Reveals Enhanced Terpenoid Biosynthesis in C. coronatus

Genome mining using antiSMASH v6.0 identified eleven biosynthetic gene clusters in *C. coronatus*, including two dedicated to terpenoid biosynthesis (Appendix A). Untargeted metabolomic profiling via LC-quadrupole time-of-flight detected 1073 metabolites in negative ion mode and 2062 metabolites in positive ion mode (Appendix A). Subsequent annotation against the KEGG and LIPID MAPS databases classified 51 metabolites as terpenoid-related compounds. The OPLS-DA model revealed 28 terpenoid-like metabolites exhibiting differential abundance between *C. coronatus* and *C. obscurus*, based on the variable importance in projection values. Of these, 22 metabolites were significantly upregulated in *C. coronatus* (Table 3), including monoterpenes, sesquiterpenes, diterpenes, tetraterpenes, and terpendoles.

## 4. Discussion

Fungi of the genus *Conidiobolus* exhibit polyphyletic development, encompassing diverse ecological niches, including soil saprotrophy, insect parasitism, plant symbiosis, and even human infection [17,18,50]. Recent studies integrating molecular phylogenetic reconstruction, morphological and ecological traits, and evolutionary trajectories have prompted the reclassification of *Conidiobolus* into multiple families, such as Neoconidiobolaceae [11]. From the perspective of insect pathogenesis, successful host infection involves three critical stages: host recognition, host invasion, and host nutrient adaptation/utilization [51]. These processes rely on activation of fungal infection machinery triggered by host-specific signals. Conversely, exposure to non-host substrates typically triggers infective conidia to produce secondary conidia for dispersal—a hallmark of entomophthoralean fungi that employ diverse spore types to facilitate transmission and survival [1,52]. Upon signal perception, fungal gene expression patterns are dramatically reconfigured via signal transduction mechanisms to adapt to heterogeneous host environments [3,53]. Genomic analysis herein revealed that *C. coronatus* possesses expanded repertoires of host recognition molecules, transcription regulators, and host-invading enzymes, thus comprehensively enabling its infection continuum (Appendix A).

Cell surface GPCRs mediate host recognitions in insect pathogens by detecting chemical/physical signals (e.g., isoflavones and hydrophobicity) and regulating infection processes [49,51,54]. Comparative genomic analysis shows that the generalist *M. anisopliae* encodes significantly more GPCR genes (61) than the specialist *M. acridum* (37). Furthermore, ablation of individual GPCR genes impairs infection structure formation, suggesting GPCRs in broad-spectrum host recognition [51,55,56]. We identified 125 GPCRs with diverse motif architectures in *C. coronatus*, suggesting a molecular basis for its broad host range.

GPCRs activate infection-related enzymes via MAPK, cAMP-PKA, or calcium signal pathways [51]. However, functional genomics in entomophthoralean fungi remain constrained by limited genetic tools. Current knowledge primarily extrapolates from model fungi. Although *C. coronatus* encodes diverse transcription factor families, functional validation is scarce. Nevertheless, many are putatively associated with pathogenicity, such as prz1 (Swiss-Prot ID), involved in regulating calcium ion homeostasis for germination [57]; TUP1, repressing pathogenesis-related genes [58]; TRY3, involved in fungal cell adherence to substrates [59]; and GIS1, involved in the Ras/cAMP pathway for nutrient adaptation [60]. Additionally, long non-coding RNAs have been reported to regulate virulence attenuation in *C*. *obscurus* [61]. Given these gaps, transcriptomic profiling of core gene clusters and non-coding regulators during *C*. *coronatus* infection is imperative.

Prior to successful colonization of the insect hemocoel, entomopathogenic fungi must overcome the host’s physicochemical barriers and evade immune defenses. The insect cuticle comprises a protein–chitin matrix embedded with lipids [62]. To breach this barrier, entomopathogenic fungi secrete cuticle-degrading enzymes (CDEs), including serine proteases (subtilisin), chitinases (GH18), and lipases [18,53]. Genomic analysis in these CDE families supports *C*. *coronatus’* capacity to invade diverse hosts. Functional annotations indicate synergistic roles; e.g., CE10 esterases remove ester-based modifications from chitin substrates to facilitate chitinase activity; protease–chitinase complexes hydrolyze cuticular matrices; and lipases degrade epicuticular waxes [18,63]. Notably, individual subtilisin contributes 19–29% to virulence in the generalist *B. bassiana* [64]. The pronounced CDE expansion in *C. coronatus* suggests exceptional cuticle penetration adaptability, underpinning its broad host range.

Meanwhile, our study has revealed numerous effector molecules that counteract diverse host immune defenses, including P450s, tyrosinases, and cysteine-rich secretory proteins (CRISPs). In *C. coronatus*, P450s, such as CYP51, facilitate fungal colonization by synthesizing ergosterol for membrane integrity and detoxify host-derived compounds [65]. Tyrosinases, copper-containing oxidases, catalyze melanin biosynthesis, which protects against the host’s hydrolytic enzymes, UV radiation, and oxidative stress by scavenging free radicals [66]. CRISPs, belonging to the CAP superfamily, disrupt host immunity through ion channel modulation and sterol binding, thereby enhancing pathogen virulence [67,68]. Additionally, *C. coronatus* produces non-ribosomal peptides, coronatin-1/2, which directly incapacitate host hemocytes by inducing cytoskeletal disassembly and apoptosis [69,70].

Host-genotype-specific variations modulate interaction outcomes. In our study, 240 PHI-annotated genes correlate with “mixed outcome” phenotypes, probably attributable to *C. coronatus*’ broad host range. Generalist pathogens target conserved host features while dynamically adapting virulence strategies to specific host genotypes [15]. The acquisition of genes encoding nonselective toxins via horizontal gene transfer (HGT) enhances pathogenic virulence [3]. For example, CytCo (a cytolytic-like gene in the aphid-obligate pathogen *C. obscurus*), probably originating from bacterial pathogens, induces toxicity in *Galleria mellonella* hemocytes and nematodes by disrupting calcium homeostasis through interacting with calcium-transporting ATPase [71,72,73,74]. Destruxins, broad-spectrum toxins targeting nerve–muscle ion channels and disrupting hemocyte-mediated immunity in diverse insects, were acquired via HGT of the dtx gene cluster from a plant-associated ancestor resembling *Alternaria* species [56]. *C. coronatus*’s metabolites also exhibit insecticidal activity across hosts, e.g., harman and norharman alkaloids exhibit efficacy against Lepidoptera hosts [75]. In this study, sesquiterpene metabolites produced via terpenoid biosynthetic gene clusters in *C. coronatus* suggest insecticidal and plant-growth-promoting effects, warranting experimental validation.

Conclusively, our integrated genomic and metabolomic analyses reveal divergent evolutionary adaptations underpinning host-range plasticity in *Conidiobolus* pathogens: *C. coronatus* (a generalist) exhibits genomic plasticity through expansions in host recognition (GPCRs), invasion (protease–CAZyme synergies), and niche exploitation (terpenoid–P450 networks), enabling broad host ranges across insect taxa, whereas *C. obscurus* (an aphid-obligate specialist) relies on focused virulence toolkits. Current limitations in genetic tools for Entomophthorales constrain functional validation of these candidates; integrating multiomic approaches will elucidate how *C. coronatus* dynamically reprograms its infectivity to adapt to diverse insect hosts. The broad-spectrum efficacy of *C. coronatus* positions it for deployment against multi-pest complexes (e.g., lepidopteran–dipteran infestations), leveraging its polyphagous adaptions for targeted biocontrol applications.

## Figures and Tables

**Figure 1 jof-11-00600-f001:**
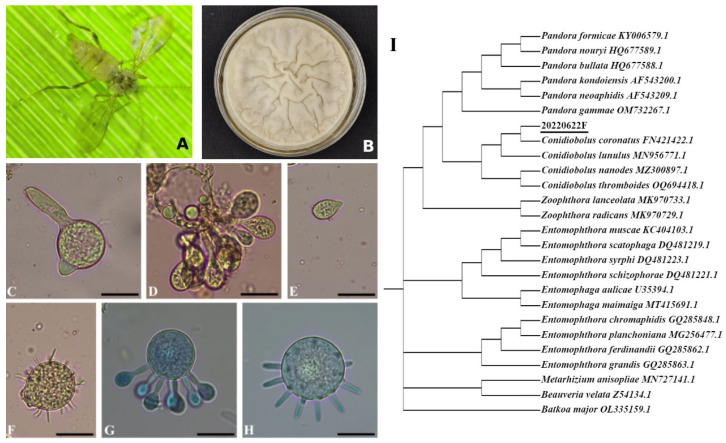
Morphological and molecular identification of *Conidiobolus* sp. strain 20220622F. (**A**) Cadaver of a naturally infected *Rhopalomyia* sp. gall midge attached to a bamboo leaf. (**B**) Purified culture on Sabouraud dextrose agar plus yeast extract in a 60 mm diameter Petri dish. (**C**–**H**) Micrographs of conidia stained with lactophenol-aceto-orcein (**C**–**F**) and cotton blue (**G**,**H**): (**C**) germinating primary conidium; (**D**,**G**,**H**) conidia at different sporulation stages; (**E**) secondary conidium; (**F**) villose conidium; (**G**,**H**) forming microconidia. (**I**) Phylogenetic analysis based on ITS sequences comparing *Conidiobolus* sp. strain 20220622F with other *Conidiobolus* species from GenBank. *Beauveria* and *Metarhizium* spp. served as the outgroup. Scale bar: 20 μm for (**C**–**H**).

**Figure 2 jof-11-00600-f002:**
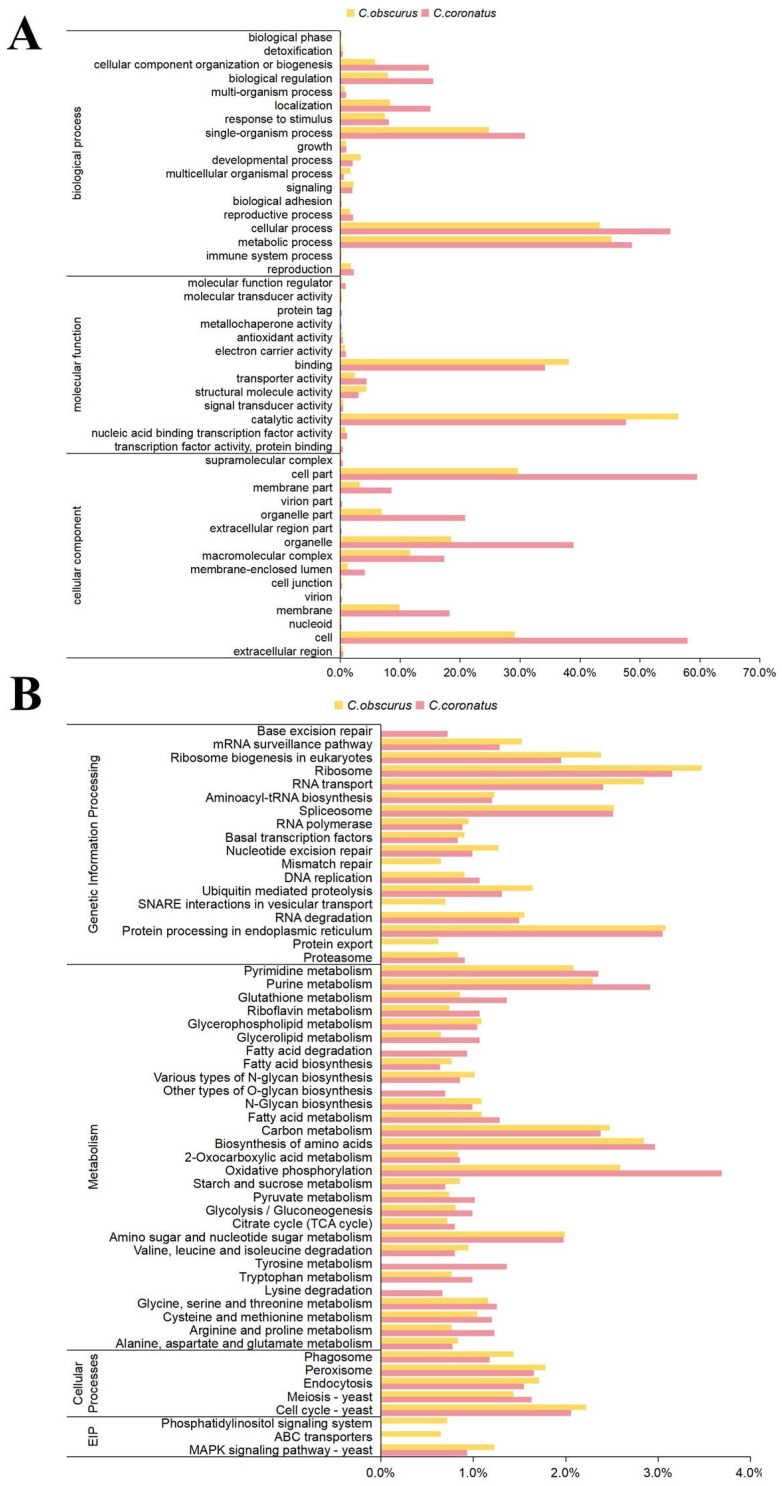
Functional divergence between the generalist *C. coronatus* and the specialist *C. obscurus*. (**A**) GO enrichment across biological process, molecular function, and cellular component categories. (**B**) KEGG pathway enrichment in genetic information processing, metabolism, and cellular processes. Bar lengths indicate the percentages of enriched genes relative to the total annotated genes. EIP: environmental information processing.

**Figure 3 jof-11-00600-f003:**
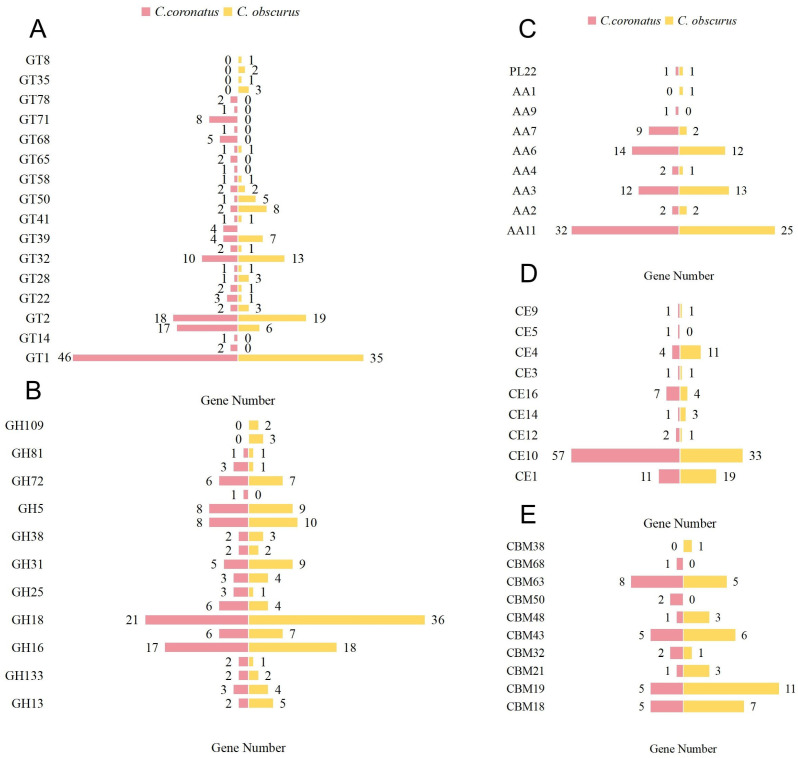
The different numbers of carbohydrate enzyme family genes in *C. coronatus* and *C. obscurus*. (**A**) Glycosyl transferases, (**B**) glycoside hydrolases, (**C**) polysaccharide lyases and auxiliary activities, (**D**) carbohydrate esterases, and (**E**) carbohydrate-binding modules.

**Figure 4 jof-11-00600-f004:**
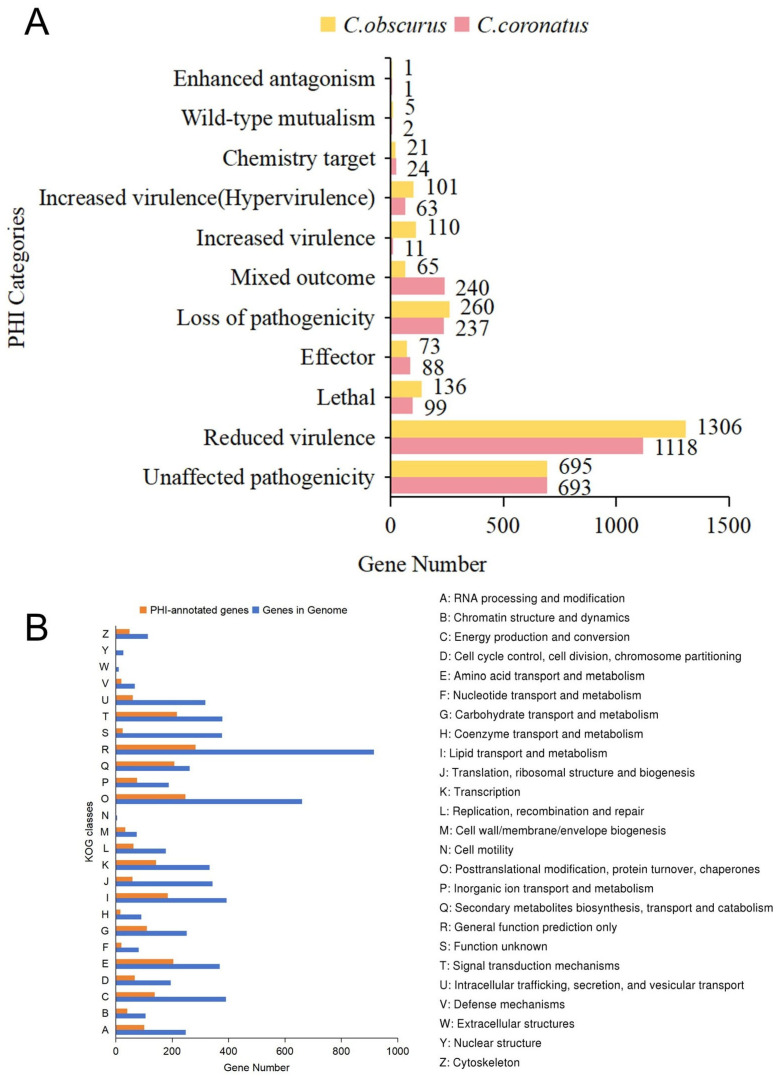
Comparative functional annotation of pathogenicity-associated genes in *C. coronatus* and *C. obscurus*. (**A**) Distribution of PHI (pathogen–host interaction) categories related to virulence phenotypes. Gene counts are shown for 11 functional categories associated with pathogenic outcomes. (**B**) Functional classification of PHI-based genes in *C. coronatus* via KOG annotation.

**Figure 5 jof-11-00600-f005:**
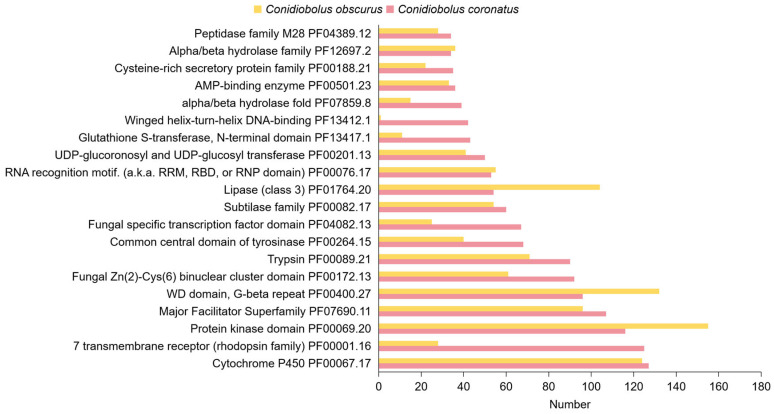
Top 20 Pfam annotation results for *C. coronatus* and *C. obscurus*.

**Figure 6 jof-11-00600-f006:**
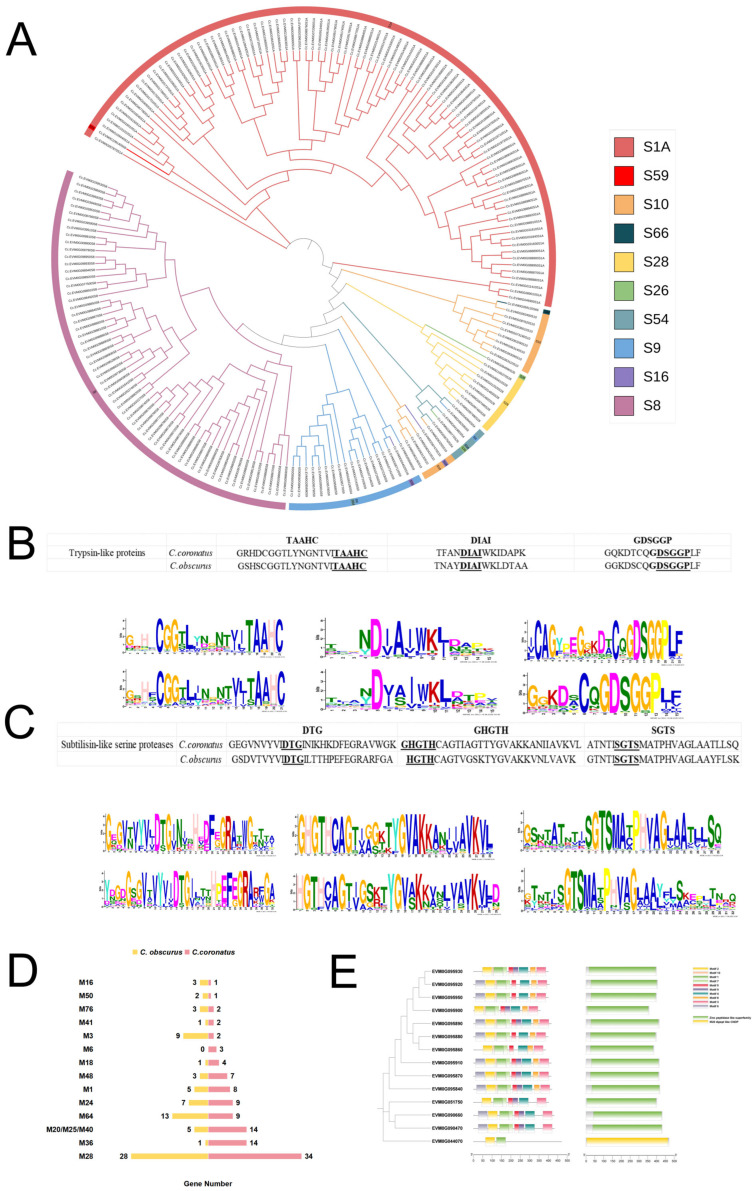
Diversity and expansion of peptidase repertoires in the *C. coronatus* genome based on Pfam annotation. (**A**–**C**) Serine protease repertoire: (**A**) Phylogenetic tree for the serine proteases in *C. coronatus*, constructed using the maximum likelihood method in MEGA v12, illustrating evolutionary relationships among these families; (**B**) conserved motifs identified in trypsins, highlighting key functional residues; (**C**) conserved motifs in subtilisins, emphasizing catalytic domains. (**D**) Comparative analysis of metalloproteinase gene families between *C. coronatus* and *C. obscurus*, showing differential expansion patterns. (**E**) Phylogenetic tree and conserved motif visualization for proteases containing the M20/M25/M40 domain (Pfam ID: PF01546.23), based on sequence alignment and domain architecture.

**Figure 7 jof-11-00600-f007:**
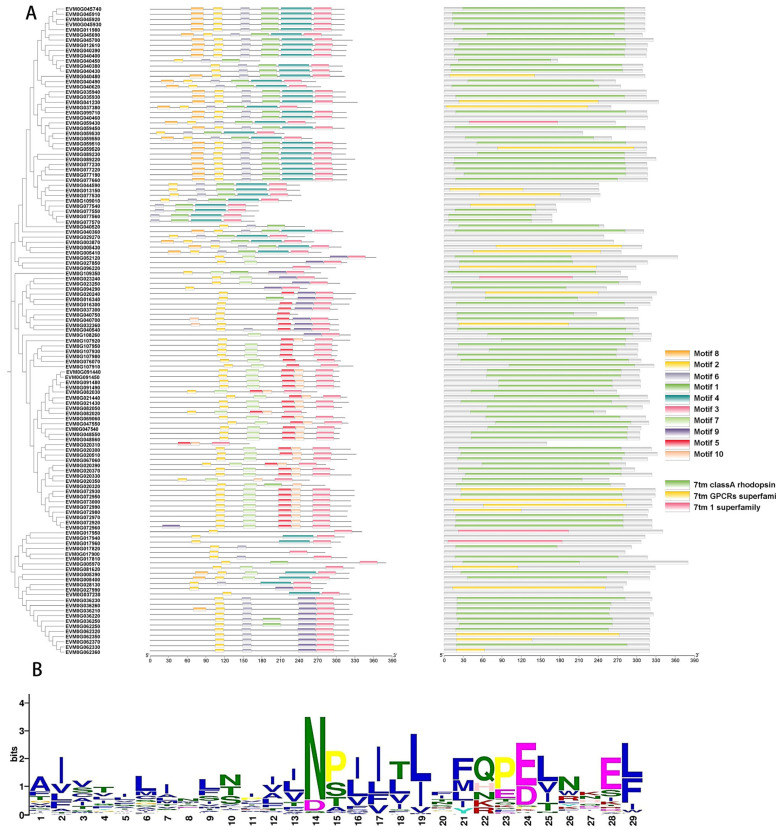
Phylogenetic classification and structural characterization of GPCRs in *Conidiobolus coronatus*. (**A**) Integrated analysis of G-protein-coupled receptors (GPCRs): (Left) Maximum likelihood phylogenetic tree of GPCR sequences, revealing evolutionary clustering patterns. (Right) Distribution of conserved structural motifs and domains across GPCR sequences. (**B**) Sequence conservation analysis of a representative motif 10 in GPCR domain.

**Figure 8 jof-11-00600-f008:**
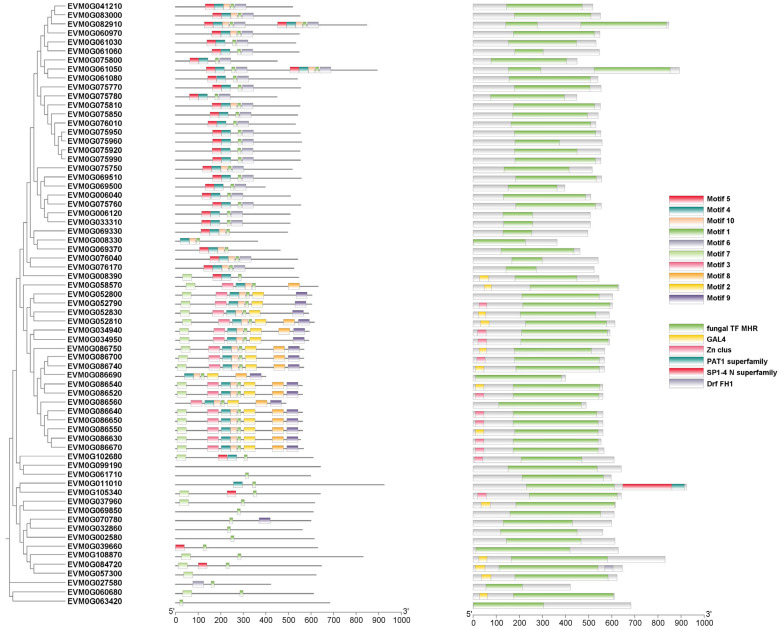
Phylogenetic classification and structural characterization of fungus-specific transcription factor in *Conidiobolus coronatus*. (**Left**) Maximum likelihood phylogenetic tree of sequences of fungus-specific transcription factor, revealing evolutionary clustering patterns. (**Right**) Distribution of conserved structural motifs and domains across these sequences.

**Table 1 jof-11-00600-t001:** Genome characteristics of *Conidiobolus coronatus* 20220622F.

Species	*C. coronatus*
Genome assembly (Mb)	44.21
Coverage (×)	82.44
Scaffold number	77
Scaffold N50 length (bp)	1,415,773
Scaffold N90 length (bp)	755,109
G + C content (%)	27.65
Protein-encoding genes	11,128
Average number of exons per gene	2.85
Repeat sequences (bp)	7,745,087
rRNA, family number	294, 4
tRNA, family number	1294, 46
Other ncRNA, family number	43, 17

**Table 2 jof-11-00600-t002:** Genomic characteristics of different entomophthoralean species.

Species [Ref.]	*C. coronatus* 20220622F	*C. coronatus* NRRL28638 [16]	*C. heterosporus* RCEF6331 [17]	*C. obscurus* ARSEF 7217 [18]	*E. muscae* ARSEF 13514 [16]
Genome size (Mb)	44.21	39.9	33.4	37.6	1030
Coverage (×)	82.44	16.31	466	318	83.12
Complete fungal BUSCOs	93.45%	~80%	/	80.0%	81.3%
Scaffolds	77	1050	96	167	7810
Scaffold N50 (bp)	1,415,773	102,410	1,660,467	1,104,530	329,600
Scaffold N90 (bp)	755,109	19,422	86,181	75,751	79,400
G + C content (%)	27.65	27.5	36.88	26.46	41.03
Repeat sequence (%)	17.52	/	12.9	17.44	90.9%
Protein-coding genes	11,128	10,568	10,857	10,262	38,917

**Table 3 jof-11-00600-t003:** Highly expressed terpene compounds detected in *C. coronatus* vs. *C. obscurus*.

Compound Name	Annotation (LIPID MAPS/KEGG)	Relative Abundance		Log_2_(FC)
		*C. coronatus*	*C. obscurus*	
Limonene-1,2-diol	C10 isoprenoids (monoterpenes) (PR0102)	658.1	286.9	1.20
(1S,4R)-1-Hydroxy-2-oxolimonene		390.1	271.5	0.52
6,10-Dimethyl-9-methylene-undec-5E-en-2-one	C15 isoprenoids (sesquiterpenes) (PR0103)	881.7	1.1	9.71
5-(1-Oxopropan-2-yl)isolongifol-5-ene		17,817.5	320.6	5.80
5-(1-Hydroxypropan-2-yl)isolongifol-5-ene		281.6	8.3	5.09
Dihydrophaseic acid		1613.9	215.5	2.90
(2-trans,6-trans)-Farnesol		1753.0	427.3	2.04
Gibberellin A8-catabolite	C20 isoprenoids (diterpenes) (PR0104)	3087.4	61.9	5.64
(-)-Reiswigin A		1965.6	144.1	3.77
Gibberellin A34-catabolite		2408.0	1187.8	1.02
(-)-2,7-Dolabelladiene-6beta,10alpha,18-triol		12,478.5	6731.4	0.89
Gibberellin A36		32,494.9	26,459.1	0.30
Astaxanthin	C40 isoprenoids (tetraterpenes) (PR0107)	2557.2	9.9	8.02
Adonixanthin		2844.9	110.6	4.69
1′-Hydroxytorulene		30,533.4	3964.0	2.95
Octaprenyl diphosphate		3803.3	1120.3	1.76
Neurosporaxanthin		1082.4	438.6	1.30
Echinenone		260.3	132.9	0.97
Thiothece-474		22,722.8	12,189.4	0.90
Canthaxanthin		841.7	499.2	0.75
Terpendole K	Indole diterpene alkaloid biosynthesis (ko00403)	177.3	50.1	1.82
Terpendole G		2762.2	1725.8	0.68

## Data Availability

The PacBio-II-assembled genome was deposited at GenBank, with accession PRJNA1264516. All the data generated or analyzed during this study are included in the published article, accessions, and Appendix A.

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
