# Peer review of "Genomic Insights into the Molecular Basis of Broad Host Adaptability of the Entomopathogenic Fungus Conidiobolus coronatus (Entomophthoromycotina)"

_jof, 2025, doi:10.3390/jof11080600_

Round 1
Reviewer 1 Report
The manuscript by Bai et al. describes the sequencing a strain of C. coronatus isolated from a Rhopalomyia sp. cadaver using PacBio long-read sequencing to elucidate the molecular basis of its wide host adaptability. This study is interesting, complete, and well written. The authors show the genomic foundations of this entomopathogenic fungus, facilitating further understanding pathogen-host interactions.
- Lines 76 and 78. It's unusual to include figures in the introduction section. I think they should be explained in the methodology and results sections. The authors mentioned that strain 20220622F strain exhibited potent pathogenicity against Hyphantria cunea larvae (Supplementary Figure S1), confirming its broad-host-range feature. However, the authors have not shown evidence of this pathogenicity in terms of a concentration-mortality relationship.
- Line 94. Please mention if C. coronatus strain 20220622F was cultured as polysporic or monosporic.
- The figures 2-8 need better quality. Some texts are very small. Consider editing it again.
- I consider that the methodology used was appropriate, but the authors should mention the origin of the C. coronatus strain 20220622F.
- For the comparative analysis between C. coronatus and C. obscurus, the authors mentioned that they performed a reannotation analysis of the genome of C. obscurus strain ARSEF 7217 using the methods described above (lines 135-136). Please, indicate whether the methods were identical to Zhang et al (2022, reference 18) or had some changes.
- In the Result section, the figures 2-8 need better quality. Some text is very small and diffuse. Consider editing these again.
- Based on the figures 2-5 and table 5, the authors have performed a comparison between both C. coronatus and C. obscurus strains related to functional divergence, the different number of carbohydrate enzyme family genes, comparative functional annotation of pathogenicity-associated genes, top 20 Pfam annotation results, and fighly expression of terpene compounds detected in these fungi. However, these comparisons have been poorly discussed, with the exception of the relationship with the long non-coding RNAs to regulate virulence attenuation in C. obscurus (lines 400 and 401) and CytCo (cytolytic-like gene, lines 436-439). Please include more detailed information about this.
- Similarly, in the conclusion paragraph (lines 447-452), the authors only mentioned the genomic and metabolomic analysis of C. coronatus, but it is necessary to include conclusions related to the comparison with the obligate pathogen C. obscurus, including its perspectives as biocontrol agents.
Author Response
- Lines 76 and 78. It's unusual to include figures in the introduction section. I think they should be explained in the methodology and results sections. The authors mentioned that strain 20220622F strain exhibited potent pathogenicity against Hyphantria cunea larvae (Supplementary Figure S1), confirming its broad-host-range feature. However, the authors have not shown evidence of this pathogenicity in terms of a concentration-mortality relationship.
Author response: We appreciate the reviewer’s feedback. The inclusion of Figure 1 (morpho-molecular identification) and Supplementary Figure S1 (pathogenicity assay) in the Introduction aimed to concisely establish the isolate’s identity and ecological relevance as a broad-host-range pathogen. This work focuses on genomic adaptations underpinning generalist, not virulence metrics. Pathogenicity assays (Supplementary Figure S1) served to phenotypically validate the strain’s broad-host-range trait. Therefore, we maintain that strain information appropriately precedes the core research sections. This placement enables readers to accurately contextualize the study subject through integrated morphological, molecular, and qualitative bioassay evidence—which collectively provide sufficient characterization. Regarding the infection dynamics of Conidiobolus coronatus against the fall webworm (Hyphantria cunea), we will prepare a comprehensive analysis presented in a separate manuscript.
2. Line 94. Please mention if C. coronatus strain 20220622F was cultured as polysporic or monosporic.
Author response: The C. coronatus strain 20220622F was isolated from a single Rhopalomyia sp. cadaver and subsequently purified through monosporic culture to ensure genetic homogeneity. We modified the sentence to make clear.
3. The figures 2-8 need better quality. Some texts are very small. Consider editing it again.
Author response: We thank the reviewer for highlighting this issue. All figures (2–8) have been revised to enhance resolution and readability. The relative data used in figures also presented in supplementary tables.
4. I consider that the methodology used was appropriate, but the authors should mention the origin of the C. coronatus strain 20220622F.
Author response: We appreciate the reviewer’s suggestion. The origin information of C. coronatus strain 20220622F is added: Zhejiang A&F University (Hangzhou, China; 30.26°N, 119.73°E).
5. For the comparative analysis between C. coronatus and C. obscurus, the authors mentioned that they performed a reannotation analysis of the genome of C. obscurus strain ARSEF 7217 using the methods described above (lines 135-136). Please, indicate whether the methods were identical to Zhang et al (2022, reference 18) or had some changes.
Author response: The core methodology for reannotating the C. obscurus genome remained consistent with Zhang et al. (2022). Key methodological extensions in this study such as GPCR annotation, Terpenoid BGC validation, etc.
6. In the Result section, the figures 2-8 need better quality. Some text is very small and diffuse. Consider editing these again.
Author response: All figures (2–8) have been revised to enhance resolution and readability.
7. Based on the figures 2-5 and table 5, the authors have performed a comparison between both C. coronatus and C. obscurus strains related to functional divergence, the different number of carbohydrate enzyme family genes, comparative functional annotation of pathogenicity-associated genes, top 20 Pfam annotation results, and fighly expression of terpene compounds detected in these fungi. However, these comparisons have been poorly discussed, with the exception of the relationship with the long non-coding RNAs to regulate virulence attenuation in C. obscurus (lines 400 and 401) and CytCo (cytolytic-like gene, lines 436-439). Please include more detailed information about this.
Author response: PHI-base annotations, CAZyme diversity, and protease functions in C. obscurus related to pathogenicity were extensively characterized in Zhang et al. (2022). This study intentionally prioritizes novel host-range determinants—particularly GPCRs, terpenoid, and protease plasticity—revealed through de novo analysis of C. coronatus. Therefore, this paper focuses exclusively on highlighting critical distinctions, avoiding a comprehensive reanalysis of CAZymes and other established functional categories in C. coronatus to preserve the analytical focus on novel host-range determinants. Meanwhile, due to the scarcity of established literature, we deliberately constrained our discussion to empirically supported observations. Expanding beyond the scope of our metabolomic and genomic data would risk speculative extrapolation, potentially introducing unsubstantiated claims. The listing of CytCo and destruxin merely serves as examples to illustrate the diversity of horizontal acquisition of exogenous gene functions by pathogens, suggesting that the terpenoid compounds discovered in C. coronatus could potentially also originate from horizontal gene transfer. The functional roles of the newly discovered genes and metabolites require further experimental validation.
8. Similarly, in the conclusion paragraph (lines 447-452), the authors only mentioned the genomic and metabolomic analysis of C. coronatus, but it is necessary to include conclusions related to the comparison with the obligate pathogen C. obscurus, including its perspectives as biocontrol agents.
Author response: We sincerely thank the reviewer for this valuable suggestion. The concluding paragraph has been comprehensively revised to integrate comparative insights with C. obscurus and explicitly address biocontrol implications.
Reviewer 2 Report
The manuscript titled "Genomic Insights into Molecular Basis of Broad Host Adaptability of the Entomopathogenic Fungus Conidiobolus coronatus (Entomophthoromycotina)" presents a thorough multi-omics study that combines long-read genome sequencing, functional gene annotation, comparative genomics, and metabolomics to explore the genetic underpinnings of C. coronatus’s broad host range. The integration of the supplementary data—particularly the extensive annotation statistics, CAZyme and protease families, secretome analysis, and metabolite profiles—adds substantial depth to the conclusions and reinforces the robustness of the findings.
Recommendation: Minor Revision
The manuscript is scientifically rigorous and data-rich, especially when considering the supplementary files. The genome and metabolome of C. coronatus are well-characterized, and the findings make a significant contribution to our understanding of fungal entomopathogenicity, host adaptation, and biocontrol potential. Only minor revisions are required to improve clarity, ensure consistency, and bolster interpretive depth.
Areas for Improvement
- Lack of Details on Strain Viability and Sporulation Rates:
While morphology and infection potential are described (e.g., Supplementary Figure S1), the authors do not quantify sporulation rate, conidial yield per plate, or mycelial biomass per flask. These are important for reproducibility, especially for pathogenicity studies. - Lack of information about the impact of environmental factors on research, I mean the fungus rearing condition mentioned in manuscript differ from conditions described in literature data, please explain
-
Experimental Validation
Despite the richness of in silico predictions, the study lacks transcriptomic, proteomic, or gene knockout data to directly validate candidate virulence factors or host-adaptation mechanisms. Given the complexity of the secretome and metabolome, targeted functional assays would significantly strengthen the claims. -
More Quantitative Pathogenicity Assessment
While Figure S1 documents the infection of Hyphantria cunea larvae, the manuscript would benefit from quantified bioassays (e.g., mortality rates, LTâ‚…â‚€/LCâ‚…â‚€) across multiple insect taxa to correlate genetic traits with observed phenotypes (at least discuss it). -
Redundancies and Formatting
There are scattered grammatical errors and some redundancy in discussion (e.g., repetition of virulence gene families). The authors should also ensure consistent formatting of gene and species names throughout. -
Gene Family Expansion Clarification
The manuscript would be strengthened by statistical analyses (e.g., CAFE) to formally test for significant expansions in gene families (e.g., CAZymes, GPCRs, TFs), rather than relying on raw counts alone. -
Minor Technical Issues in Supplement
Some supplementary tables lack consistent units or headings (e.g., Table S1 abbreviations). These should be clarified for reproducibility and accessibility.
Minor Suggestions:
-
Add a schematic summarizing key expanded gene families and their functional roles across infection stages.
-
Include additional host-range metadata, if available, from lab or field studies.
-
Provide a summary table linking major findings to ecological implications (e.g., CAZyme → cuticle degradation → host penetration).
Author Response
Minor comments:
- Lack of Details on Strain Viability and Sporulation Rates:
While morphology and infection potential are described (e.g., Supplementary Figure S1), the authors do not quantify sporulation rate, conidial yield per plate, or mycelial biomass per flask. These are important for reproducibility, especially for pathogenicity studies.
Author response: Supplementary Figure S1 (pathogenicity assay) in the Introduction aimed to indicate the isolate characteristics as a broad-host-range pathogen. This work focuses on genomic adaptations underpinning generalist, not virulence metrics. Regarding the detailed infection dynamics of Conidiobolus coronatus against the fall webworm (Hyphantria cunea), we will prepare a comprehensive analysis on another manuscript of virulence assay and host-pathogen interaction based on transcriptomics.
- Lack of information about the impact of environmental factors on research, I mean the fungus rearing condition mentioned in manuscript differ from conditions described in literature data, please explain
Author response: We appreciate the reviewer’s insightful observation regarding fungal rearing conditions. Our cultivation protocol for C. coronatus was adapted from established methods for C. obscurus, as both species share conserved physiological requirements within the Conidiobolus genus. Crucially, the DNA extraction and genomic analysis procedures remain unaffected by these rearing conditions, as nucleic acid integrity and subsequent sequencing fidelity are maintained through standardized isolation protocols and quality control measures.
- Experimental Validation
Despite the richness of in silico predictions, the study lacks transcriptomic, proteomic, or gene knockout data to directly validate candidate virulence factors or host-adaptation mechanisms. Given the complexity of the secretome and metabolome, targeted functional assays would significantly strengthen the claims.
Author response: We appreciate the reviewer's valuable suggestion regarding experimental validation. While gene knockout approaches would indeed strengthen mechanistic insights, these techniques currently face methodological constraints inherent to functional validation in Entomophthorales fungi. The proposed transcriptomic and proteomic analyses represent important future directions that will be addressed in a subsequent manuscript focused on host-pathogen interaction dynamics, distinct from the genomic focus of the present study.
More Quantitative Pathogenicity Assessment
While Figure S1 documents the infection of Hyphantria cunea larvae, the manuscript would benefit from quantified bioassays (e.g., mortality rates, LTâ‚…â‚€/LCâ‚…â‚€) across multiple insect taxa to correlate genetic traits with observed phenotypes (at least discuss it).
Author response: As indicated in our earlier response, virulence assays of C. coronatus against H. cunea larvae fall beyond the genomic scope of this study. These experimental validations will be presented in a separate manuscript focused specifically on host-pathogen interaction dynamics.
- Redundancies and Formatting
There are scattered grammatical errors and some redundancy in discussion (e.g., repetition of virulence gene families). The authors should also ensure consistent formatting of gene and species names throughout.
Author response: We thank the reviewer for highlighting these important editorial considerations. We make necessary modifications.
- Gene Family Expansion Clarification
The manuscript would be strengthened by statistical analyses (e.g., CAFE) to formally test for significant expansions in gene families (e.g., CAZymes, GPCRs, TFs), rather than relying on raw counts alone.
Author response: We appreciate the reviewer's insightful suggestion regarding statistical validation of gene family expansions. Our current comparative approach prioritizes direct quantitative comparison of their genomes traits on host ranges: Comparable genomic scales: C. coronatus (11,128 annotated genes) and C. obscurus (10,568 genes) exhibit nearly identical total gene counts, making raw count comparisons biologically meaningful for initial expansion assessment. Accessibility focus: Direct numerical presentation (e.g., GPCRs: 125 vs. 28; CAZymes: 430 vs. 367) provides immediate intuitive understanding of functional differences for broad readership.
- Minor Technical Issues in Supplement
Some supplementary tables lack consistent units or headings (e.g., Table S1 abbreviations). These should be clarified for reproducibility and accessibility.
Author response: modified.
Minor Suggestions:
- Add a schematic summarizing key expanded gene families and their functional roles across infection stages.
Author response: We have added Supplementary Table S19 to comprehensively summarize the expanded gene families and their functional roles across infection stages, as requested.
- Include additional host-range metadata, if available, from lab or field studies.
Author response: sorry, unavailable data.
- Provide a summary table linking major findings to ecological implications (e.g., CAZyme → cuticle degradation → host penetration).
Author response: We appreciate the suggestion to summarize gene family functions in tabular form. We presented the information in the supplementary Table S19.
Round 2
Reviewer 1 Report
I have no major comments
I appreciate the opportunity to review again the MS jof-3761496. The authors have modified and improved the manuscript that was submitted previously. I only have a minor correction: It is not necessary to mention the supplementary table (line 435) in the conclusion.
Author Response
Minor comments:
I appreciate the opportunity to review again the MS jof-3761496. The authors have modified and improved the manuscript that was submitted previously. I only have a minor correction: It is not necessary to mention the supplementary table (line 435) in the conclusion.
Author response: Thanks for your suggestion! we transfer (Supplementary Table S19) to the first paragraph in Discussion.